# A Utility Framework for COVID-19 Online Forward Triage Tools: A Swiss Telehealth Case Study

**DOI:** 10.3390/ijerph19095184

**Published:** 2022-04-24

**Authors:** Janet Michel, Annette Mettler, Martin Müller, Wolf E. Hautz, Thomas C. Sauter

**Affiliations:** Department of Emergency Medicine, Inselspital, University Hospital, University of Bern, 3010 Bern, Switzerland; annette.mettler@insel.ch (A.M.); martin.mueller2@insel.ch (M.M.); wolf.hautz@insel.ch (W.E.H.); thomas.sauter@insel.ch (T.C.S.)

**Keywords:** COVID-19, evaluation framework, online forward triage tools, telehealth, utility

## Abstract

The SARS-CoV-2 pandemic caused a surge in online tools commonly known as symptom checkers. The purpose of these symptom checkers was mostly to reduce the health system burden by providing worried people with testing criteria, where to test and how to self-care. Technical, usability and organizational challenges with regard to online forward triage tools have also been reported. Very few of these online forward triage tools have been evaluated. Evidence for decision frameworks may be of particular value in a pandemic setting where time frames are restricted, uncertainties are ubiquitous and the evidence base is changing rapidly. The objective was to develop a framework to evaluate the utility of COVID-19 online forward triage tools. The development of the online forward triage tool utility framework was conducted in three phases. The process was guided by the socio-ecological framework for adherence that states that patient (individual), societal and broader structural factors affect adherence to the tool. In a further step, pragmatic incorporation of themes on the utility of online forward triage tools that emerged from our study as well as from the literature was performed. Seven criteria emerged; tool accessibility, reliability as an information source, medical decision-making aid, allaying fear and anxiety, health system burden reduction, onward forward transmission reduction and systems thinking (usefulness in capacity building, planning and resource allocation, e.g., tests and personal protective equipment). This framework is intended to be a starting point and a generic tool that can be adapted to other online forward triage tools beyond COVID-19. A COVID-19 online forward triage tool meeting all seven criteria can be regarded as fit for purpose. How useful an OFTT is depends on its context and purpose.

## 1. Introduction

The SARS-CoV-2 pandemic caused a surge in online tools, commonly known as symptom checkers [1,2]. Looking at history, pandemics have killed as many people as all wars combined [3], but many countries, if not all, were not well prepared for the SARS-CoV-2 pandemic [3].

Faced with unprecedented volumes of calls from worried and concerned citizens, many hospitals put together teams that developed these symptom checkers or online forward triage tools (OFTT) [1,4]. The purpose of these symptom checkers was mostly to reduce the health system burden by providing worried people with testing criteria and where to test. Many technological innovations have been made to support communication and patient care during the pandemic. One systematic review revealed that despite a boom in symptom checkers, user engagement was lacking throughout the design process of such online tools [5]. Technical, usability and organizational challenges have also been reported [5]. There is a need for user-centered design approaches in OFTT development (system design stage) that take into account three socio-technical factors, namely cognitive and physical stressors, workflow and context [5]. The application and effectiveness of these OFTTs depend on how usability issues and human factors in technology are resolved.

Research has revealed that many people across all age groups, including the elderly, have used the COVID-19 OFFTT. A total of 85% of the people followed the recommendations given by the online tool, demonstrating its utility in supporting medical decision-making [6,7]. Different types of telemedicine technology exist, among others, ambulance-based, wearable, handheld and the internet of things (IoT) devices [5]. One of the goals of informed decision-making is to provide individuals and families with understandable, accurate and balanced information to make a medical decision, e.g., screening, treatment and other health-related options. 

Telemedicine has been shown to improve shared decision-making, thereby increasing patient participation in medical decision-making [8,9,10]. When compared to health care professionals, OFTTs have been shown to be risk-averse [11]. SARS-CoV-2 presented additional issues in medical decision-making under time strains and decision-making under uncertainty due to limited data [12,13]. 

### The Intervention OFTT

Our OFTT was developed by the working group e-emergency medicine at the emergency department (ED) Inselspital University Hospital Bern, together with the Department of Infectious diseases, Inselspital University Hospital Bern. All participants aged 18 and above that used the OFTT between 2 March and 12 May 2020 were included. COVID-19 recommendations in Switzerland frequently changed during this time period. COVID-19 testing reagents and capacity were a challenge, as well as the increased risk of a health system overload. In this pandemic phase, the Federal Office of Public Health (FOPH) recommended testing only for symptomatic patients after travel to high-risk countries (e.g., Italy and China) or symptomatic contacts of coronavirus patients. A few weeks later (from the 20 March 2020), the recommendation was adapted to include the testing of high-risk groups (older than 65 years, pre-existing conditions and healthcare workers). Based on how the virus spread and the availability of the testing capacity, the countries and risk groups were regularly adjusted. The virus spread rapidly, and a universal test recommendation was made by the Federal Office of Public Health (FOPH)- on 27 April 2020. All symptomatic individuals were eligible to be tested. With this recommendation, our OFTT became obsolete and was finally removed from the website on 12 May 2020, paving the way for a second-generation OFTT.

When developing frameworks, the diverse needs stakeholders have should be met, and the recommendations need to be relevant to all fair-minded parties [12]. A theory or framework provides a road map for systematically identifying factors that affect implementation [14].

Evidence for decision frameworks comprise criteria and procedural guidance that ensures that all relevant factors are considered, and the underlying rationale is transparent [12]. Evidence for decision frameworks may be of particular value in a pandemic setting where time frames are restricted, uncertainties are ubiquitous, and the evidence base is changing rapidly [12]. Many OFTTs were set up primarily to reduce the health system burden [11]. There is a paucity of data when it comes to the usefulness of these OFTTs to policy implementers, policymakers and end-users. Despite the OFTT boom, very few OFTTs have been evaluated [12]. One of the reasons could be a lack of evaluative frameworks. To the best of our knowledge, there is no existing framework to assess the utility of COVID-19 OFTTs. The purpose of this manuscript is to present and propose an evaluative COVID-19 OFTT utility framework as the pandemic lingers. 

## 2. Materials and Methods

The study is embedded in a broader multi-phase explanatory sequential mixed-methods study conducted by the department of emergency telemedicine of the University of Bern. The OFTT was developed by the working group e-emergency medicine at the emergency department (ED) Inselspital University Hospital Bern, together with the Department of Infectious diseases, Inselspital University Hospital Bern. Quantitative data was collected from OFTT users between 2 March and 12 May 2020 from participants aged 18 and older that consented to the study. Qualitative data was collected in September 2020. Detailed quantitative and qualitative data from this study are currently under review elsewhere [15].

### 2.1. Framework Development

The development of the OFTT utility framework was conducted in three phases. The process was guided by the socio-ecological framework for adherence that states that patient (individual), societal and broader structural factors affect adherence to the tool [16,17]. The “best fit” synthesis technique and the WHO-INTEGRATE COVID-19 framework guided our criteria development [12]. In a further step, pragmatic incorporation of themes on the utility of OFTTs that emerged from our study [6] as well as from the literature [18,19]. See Table 1.

#### 2.1.1. Phase 1: OFTT Set Up

A COVID-19 OFTT was developed and set up by the Insel Emergency team. The hypothesis was that an OFTT can reduce the health system burden by triaging patients, providing information on symptoms and how to self-care, and directing them to the testing centers or the next level of care. Quantitative data of users were collected (n = 6272 users visited the OFTT). See Table 1.

#### 2.1.2. Phase 2: Quantitative Data Collection

The database complied with Swiss laws on personal health-related information collection. To minimise the barrier to OFTT use, as well as for legal data protection issues, no personal data were collected within the OFTT. Further data on OFTT users were collected in a second step from participants who gave explicit consent and provided email addresses to be contacted. A total of 560 participants consented to a follow-up survey and provided a valid email address and filled in an online questionnaire (22 questions). The quantitative data were analysed in Stata^®^ 16.1 (StataCorp, The College Station, TX, USA). Descriptive statistics for all variables as mean and standard deviation, median and interquartile range or frequency were determined by the type and distribution of the data. The manuscript is currently under review [6]. See Table 1.

#### 2.1.3. Phase 3: Qualitative Data Collection 

To better understand and explain the quantitative results, interviews were held with participants that further consented to a follow-up qualitative study. Video, rather than face-to-face, interviews were held in September 2020. This was a precautionary measure to protect both researchers and participants. The participants included patients (n = 19) and health care providers and health authorities (n = 5). To ensure inclusiveness in the patient group, participants of all age groups (purposeful and quota sampling) were included. One manuscript has been published [20]. See Table 1.

*Measures to ensure trustworthiness of data:* Dependability was ensured through iterative data collection and analysis and continuously adjusting our interview guide to capture newly emerging themes. Two qualitative researchers kept reflexive journals throughout the study, and debriefings were held at the end of each interview. A thick description of participants, context and the data collection process has been outlined to ensure transferability. The data were managed and analysed with the aid of MAXQDA2018.

#### 2.1.4. Phase 4: Brainstorming, Triangulation and Discussion

Brainstorming and discussions were held with the research team to develop a generic framework based on evidence from the literature and results and findings from our OFTT study. The input from the stakeholders, patients, health care providers and health authorities on their personal OFTT experience, the purpose and expectations and recommendations guided the process. We finalized by critically reflecting the criteria and what it covers, triangulating with the results and findings, see Table 2. The “best fit” synthesis technique and the WHO-INTEGRATE COVID-19 framework guided our framework development [12].

### 2.2. Stakeholders

We found the following three key stakeholders’ input as important in determining what the utility attributes of a COVID-19 OFTT are
Patients: the direct users of OFTTsHealth care providers, doctors and GPs: they know the questions that are being asked by patients and help define the essential contents of an OFTTHealth care authorities: They are responsible for health care provision and legal frameworks and criteria to be considered. See Figure 1.

### 2.3. Ethics

The local ethics committee of the Canton of Bern, Switzerland, deemed this project a quality evaluation study and waived the need for a full ethical review (Req-2020-00289) on 23 March 2020. 

## 3. Results

### How to Apply the Framework

This framework is intended to be a starting point and a generic tool that can be adapted to other OFTTS beyond COVID-19. The framework development was guided by the “best fit” synthesis technique and the WHO-INTEGRATE COVID-19 framework [12].

The seven criteria are; (i) tool accessibility, (ii) the tool as a reliable source of information, (iii) the tool’s utility in assisting with medical decision-making, (iv) the tool’s potential to allay fear and anxiety, (v) the tool’s potential to reduce the health system burden, (vi) the tool’s potential to reduce cross-infection and (vii) the tool’s utility in capacity building, planning and resource allocation-systems thinking.

The seven criteria can be adapted to suit a specific purpose, e.g., the addition of more criteria to suit the purpose of OFTT, e.g., in acute or chronic conditions. See Figure 2. How useful an OFTT is can be assessed by how the tool meets the seven criteria. A COVID-19 tool meeting all seven criteria is fit for purpose. A COVID-19 tool meeting five to six criteria is good, and a COVID-19 tool meeting four or less criteria is insufficient. The number of criteria is determined by the purpose of the OFTT. A chronic pain OFTT might have more or less criteria as compared to a urinary tract infection OFTT. An OFTT whose main purpose is to provide information might need to fulfil criteria 1 and 2, tool accessibility and reliable and comprehensive information source. There is no one size fits all. The identified criteria are meant primarily for COVID-19 OFTTs. The context and purpose of the tool will dictate the relevant criteria to be fulfilled. See Table 3.

## 4. Discussion

We propose the developed COVID-19 OFTT utility framework as a starting point and a raw guide for the development of OFTT utility frameworks in this and future pandemics. The seven criteria are; (i) tool accessibility, (ii) the tool as a reliable source of information, (iii) the tool’s utility in assisting with medical decision-making, (iv) the tool’s potential to allay fear and anxiety, (v) the tool’s potential to reduce health system burden, (vi) the tool’s potential to reduce cross-infection and (vii) the tool’s utility in capacity building, planning and resource allocation systems thinking. See Figure 2. 

The use of OFTTs is evolving, and this framework can be adapted to meet the needs of other conditions beyond COVID-19. See Table 3. The views of the main stakeholders in a health system, patients, health care providers and health authorities, culminated in the identified criteria. The development of the OFTT utility framework was a three-phase process. The multi-phase explanatory sequential mixed methods study design enabled the team to make use of both quantitative and qualitative data, thereby providing a holistic picture. The socio-ecological framework for adherence states that patient (individual), societal and broader structural factors affect adherence to a tool (*Figure 1 Social Ecological Framework for Nutrition and Physical Activity…*; *WHO|The ecological framework*.). The involvement of patients, health care providers and authorities enabled researchers to explore and incorporate individual, acceptability and equity considerations, societal, implementation, feasibility and resource implications [12,16].

The “best fit” synthesis technique and the WHO-INTEGRATE COVID-19 framework guided our criteria development [12]. In the final step, pragmatic incorporation of themes on the utility of OFTTs that emerged from our quantitative and qualitative studies (*Effects and Utility of an Online Forward Triage Tool during the SARS-CoV-2 Pandemic*.) as well as from the literature [18,19] was performed. 

Our study revealed that the OFTT has the potential to reduce the health system burden, forward transmission reduction, assist in medical decision-making and can act as a reliable information source concurring with studies elsewhere [22]. Access and socio-technical issues need to be taken into account at the point of OFTT system design, and this can only be done through the involvement of all stakeholders [5]. Coronatest.ch was one of the first COVID-19 OFTTs in Switzerland. Despite the boom of OFTTs during the pandemic, there is a paucity of evaluative research on prehospital communication technology such as OFTTs [5]. 

All stakeholders highlighted the need for the tool to be accessible, to be a reliable source of information, to assist in medical decision-making and to allay fear and anxiety to a certain extent since not all fear can be addressed. The potential of the tool in reducing the health system burden and forward transmission, as well as utility in capacity building, planning and resource allocation, e.g., tests and PPE-systems thinking, was demonstrated by our data as well as cited by the key stakeholders. Guided by the “best fit” synthesis technique and the WHO-INTEGRATE COVID-19 framework, the seven criteria for our OFTT framework criteria were developed [12]. See Table 3 and Figure 2. We have tested the framework as an analytic lens in a separate but related study to evaluate the utility of a child-specific COVID-19 OFTT, coronabambini. The seven criteria were successfully replicated and supported by evidence and quotes from parents and guardians that used the tool. The manuscripts are currently under review at Frontiers in Public Health [23]. We, therefore, encourage other researchers to test the framework in other contexts.

### Strengths and Limitations

We employed a mixed study design to gain a holistic picture of OFTT issues, challenges and what the patients, health care providers and health authorities want when they use an OFTT [13]. 

One shortcoming might be our sample size from the survey. Only a limited number of OFTT users took part in the study. This bias is due to the data protection regulations that impose voluntary participation and prohibits the automatic tracking of participants. The data source triangulation health care workers, health care providers and health authorities gave us a rounded and balanced view of what a COVID-19 OFTT ought to cover from the perspectives of these different but equally important stakeholders in the health system. Our framework is grounded and informed by real evidence from a mixed study that was embedded in one of the first COVID-19 OFTTs set up in Switzerland. This being one of the first COVID-19 OFTT utility frameworks, more studies are needed as this framework might not be suitable for other contexts and settings.

## 5. Conclusions

COVID-19 data is evolving very fast. The insights gained in our study might become obsolete as new knowledge emerges. This was one of the first COVID-19 OFTTs set up in Switzerland. The framework provides a road map for systematically identifying factors that affect implementation, in this instance, COVID-19 OFTT implementation. A telemedicine monitoring framework to assist infected patients remotely was developed elsewhere [24]. The authors argue that telemedicine has the potential to help healthcare sources in pandemics [24]. There is very little research published on the utility of a COVID-19 OFTT beyond health system burden reduction. We propose this framework as a starting point. As more data becomes available, this framework needs to be updated and adapted according to the setting and context.

## Figures and Tables

**Figure 1 ijerph-19-05184-f001:**
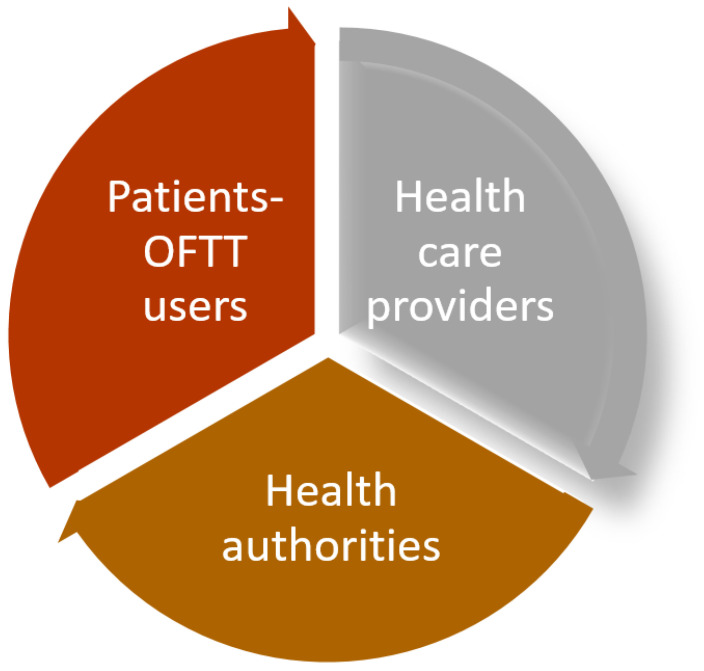
Stakeholders.

**Figure 2 ijerph-19-05184-f002:**
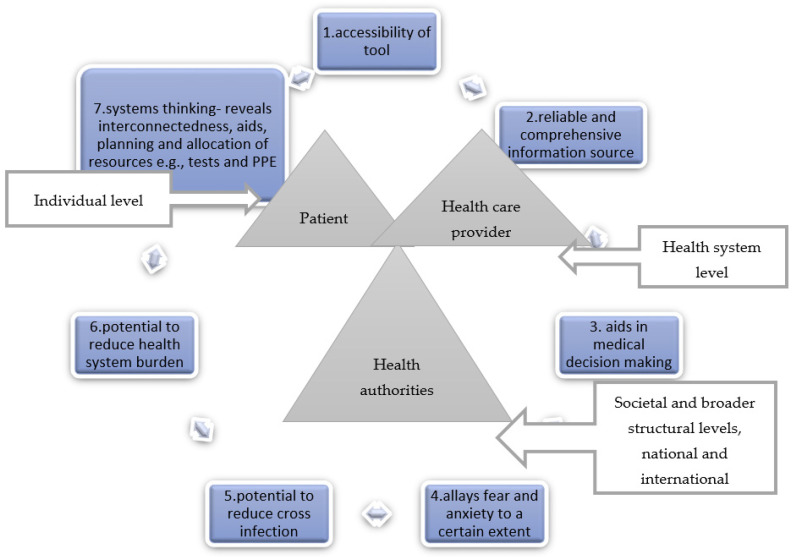
COVID-19 OFTT Utility Framework.

**Table 1 ijerph-19-05184-t001:** OFTT stages.

Stage 1: Building Up and Setting Up OFTT, Vetting the Engine, Algorithms etc.	Stage 2: Ensuring Content Validity, Reliability and Comprehensiveness and Timeliness-Evidence Constantly Changing in Novel Conditions Science vs. Social Media News	Stage 3. Effects on Individual, Societal and Health System and Broader Structural Levels
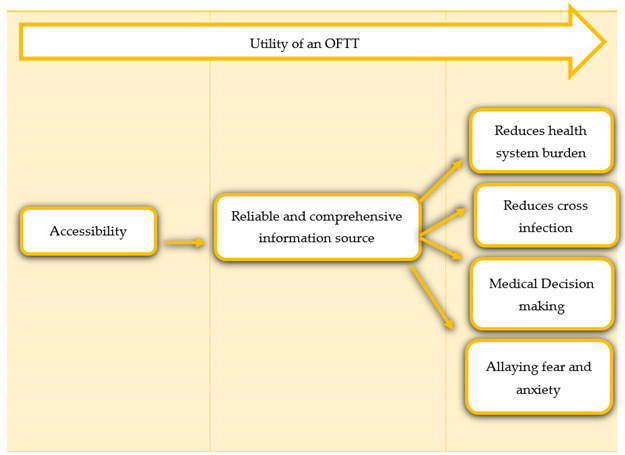		

**Table 2 ijerph-19-05184-t002:** Data sources and implications for an OFTT.

Stage and Type of Data	Results	Implications
**Online OFTT set up**	Number of completed requests between 2 March and 12 May 2020 (n = 6272) [6]	The tool was not advertised, despite that, many people visited the tool-demonstrating that the Swiss society has embraced OFTTs
**Survey**	The mean age of the participants of the email survey was 50.1 years (SD 15.4, range 18–82), with 57.4% being female	All age groups accessed the OFTT
	87.5% of the survey participants found the website helpful [6]	Participants trusted and followed the tool recommendations
	85.2% of the participants followed the recommendations given to test or not test [6].	85% (*Effects and Utility of an Online Forward Triage Tool during the SARS-CoV-2 Pandemic*.)
	65% of the participants called their GP ahead of their visit [6]	Participants followed the recommendation to call the health care provider ahead of time in case they suspected COVID-19 (*Effects and Utility of an Online Forward Triage Tool during the SARS-CoV-2 Pandemic.*)
	41.1% of all users would have contacted the GP or visited a hospital had the tool not existed; furthermore, 16.8% would have contacted a hotline (*Effects and Utility of an Online Forward Triage Tool during the SARS-CoV-2 Pandemic.*)	Many participants would have called the GP or hotline, so the tool potentially reduced the burden on the health system (*Effects and Utility of an Online Forward Triage Tool during the SARS-CoV-2 Pandemic.*)
**Face-to-face interviews**	Themes that emerged accessibility, utility of tool in preventing cross-infection, utility of tool in reducing health system burden, utility of tool as a comprehensive and reliable information source, utility of tool in allaying fear and anxiety and utility of tool in medical decision-making	Our main qualitative findings demonstrated that a COVID-19 OFTT can go beyond reducing the health system burden. Our primary hypothesis to having the potential to prevent cross-infection, allaying fear and anxiety to a certain extent, facilitating decision-making and providing reliable information, once accessibility issues are overcome These themes and concepts guided our framework

**Table 3 ijerph-19-05184-t003:** COVID-19 OFTT Criteria.

Criteria	Stakeholder	Description of the Criteria	Points
Accessibility of tool	Patient Health care provider Health Authorities	This covers the implication that tool accessibility can be assessed by how high up the tool appears on top search engines, such as Google. This also covers the implication that the language is clear, the length of the tool, i.e., how long it takes a sick patient to respond to the questions, including how long it takes for the patient to get the recommendation to call 911, completion rates, number of questions, etc., all affect tool accessibility [21]	1
A reliable source of information	Patient Health care provider Health Authorities	This covers the implication that how comprehensive a tool is depends upon the users finding the information that led them to the search engine, for example, with regards to symptoms, when to test, where to test, including contact numbers, when to call a doctor and how-to self-care	1
Medical decision-making	Patient Health care provider Health Authorities	This covers the implication that the utility of a tool can be measured by the proportion of users who follow the recommendations given by the tool. This covers the implication that trust in the tool is very important. Transparency with regards to tool validation and algorithm behind recommendations increases trust, e.g., you have a 95% probability of having COVID-19, test, or you have a 5% probability of having COVID-19, do not test	1
Allaying fear and anxiety	Patient Health care provider Health Authorities	This covers the implications of feeling better, reassured and/or less anxious after filling the OFTT. Knowledge is power	1
Reducing health system burden	Patient Health care provider Health Authorities	This covers the implication of being able to access health care services and or knowledge within the comfort of one’s own home without the need to visit a healthcare facility	1
Reducing cross-infection	Patient Health care provider Health Authorities	This covers the implication of communicating with the health care providers well ahead of visiting and making them aware a suspect is visiting the practice and taking all necessary precautions and arrangements	1
Systems thinking-showing interconnectedness-	Patient Health care provider Health Authorities	This covers the implications that a health system is a complex, adapted system. Issues affecting one area, e.g., supply chain, will affect other areas. This covers the implication that the utility of the tool is dependent upon other health systems and societal components, e.g., participants were told by the tool to test only to be told that there are no tests (shortages). This covers the implication that economic factors, such as the cost of the test influenced some not to test This covers the implication that a new life-threatening disease in a population is associated with psycho-social and behavioural issues that need to be taken into account. Fear of a positive test, psychological readiness, can prevent people from testing	1
Total points			7 points =OFTT is fit for purpose, i.e., COVID-19. 5–6 points =OFTT is good but can be improved. 4 or less points =OFTT is insufficient. Criterion based utility The usefulness of a tool can also be based on criteria fulfilled. Depending on context and purpose, the more criterion fulfilled, the better the tool For example The tool is useful in reducing the health system burden and as an information source

## Data Availability

Interviews cannot be shared openly since the participants were assured of anonymity and therefore cannot be shared publicly. Any requests can be sent to the corresponding author.

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
