# Peer review of "A Utility Framework for COVID-19 Online Forward Triage Tools: A Swiss Telehealth Case Study"

_ijerph, 2022, doi:10.3390/ijerph19095184_

Round 1
Reviewer 1 Report
Thank you for submitting the manuscript to this journal. You have chosen an interesting subject to do this study. Some issues need to be clarified. I have summarized these concerns below: the following comments need to be taken with care in order to improve the quality of the manuscript for publishing.
ABSTRACT
The elaboration of the abstract does not adapt to the recommendations of the journal, the subsections (Objectives, methods...) must be eliminated. The abstract contains abbreviations these must be eliminated, the abbreviations must appear in the introduction section, indicating the full name and then the abbreviation. On the other hand, the objective needs to be clarified and appear more explicitly in the abstract.
KEYWORDS
Keywords must be in alphabetical order.
Keywords should be reviewed, so that the manuscript can be located more easily there are concepts such as telehealth that have not been included, search for MeSH descriptors.
INTRODUCTION
Line 37. They could be more explicit and indicate the technological innovations that have been developed during the pandemic and support it with previous research.
Another important aspect that should be improved is that the introduction should be well organized and go from the general to the specific.
If the authors address this triage system, it would be interesting for them to delve into what has been done up to now.
Specify the resources and tools available.
Finally, highlight the novelty of this publication since it is not clear.
METHODS
This section is incomplete the following subsections are missing:
In phase two, it is not clear if these participants have been chosen through convenience sampling, there is no calculation of the sample size, nor inclusion and exclusion criteria.
The authors do not indicate any guidelines on which they have relied for the development of the manuscript in order to guarantee its quality.
Likewise, they do not provide references that support that the steps followed are adequate, this suggests that the manuscript may not have sufficient quality to be published.
The authors explicitly self-cite in the discussion, please ignore that information.
In phase 3, if it is included in the manuscript, there are aspects that are not clear, such as interviewed participants, how the interviews were carried out, where they were carried out, how the data analysis was carried out.
DISCUSSION
Avoid the first person plural in the wording.
The discussion of the manuscript is very short, no contract with previous research is observed. The results obtained must be contrasted with previous studies and there is only one reference in this section.
The authors should carry out a deeper discussion.
It would be necessary to include new future lines to investigate, after the completion of the study.
It would be convenient to indicate the implications of the manuscript.
CONCLUSION
Check the wording, avoid the use of the first person plural.
Line 176: The insights gained in our study might become obsolete as new knowledge emerges. This makes me think, if the data has been collected in 2020. And we are in 2022, it has been checked if it is obsolete.
Another doubt that arises to me is that if after the elaboration of the framework it has been applied in clinical practice, that is to say, it has been piloted.
REFERENCES
References must be adapted to the format of the journal.
Author Response
Inselspital
Freiburgstrasse 18
3010 Bern
08 April 2022
Editor IJERPH
Dear Editor
RE: A Utility Framework for COVID-19 Online Forward Triage Tools: A Swiss Telehealth case study
We thank the reviewers for their invaluable comments on our manuscript; “A Utility Framework for COVID-19 Online Forward Triage Tools: A Swiss Telehealth case study.” We have carefully taken all the reviewers’ comments into account and are pleased to submit a revised version herewith. We indicate below how we have changed the manuscript and respond to the reviewers’ comments item by item.
Sincerely,
Janet Michel, RN, BA Cur, MPH, PHD
Annette Mettler, MD
Thomas C. Sauter, MD, MME, Prof
Reviewer 1
Reviewer comment 1: Thank you for submitting the manuscript to this journal. You have chosen an interesting subject to do this study. Some issues need to be clarified. I have summarized these concerns below: the following comments need to be taken with care in order to improve the quality of the manuscript for publishing. ABSTRACT: The elaboration of the abstract does not adapt to the recommendations of the journal, the subsections (Objectives, methods...) must be eliminated. The abstract contains abbreviations these must be eliminated, the abbreviations must appear in the introduction section, indicating the full name and then the abbreviation. On the other hand, the objective needs to be clarified and appear more explicitly in the abstract.
*Author response
Thank you for these comments. We have adapted the Abstract, removed all abbreviation and clarified the objectives. See lines 31 to 53 now read as follows;
Abstract
Introduction: The SARS-COV-2 pandemic caused a surge in online tools commonly known as symptom checkers. The purpose of these symptom checkers was mostly to reduce the health system burden, by providing worried people with testing criteria, where to test and how to self-care. Technical, usability and organizational challenges with regards to online forward triage tools have also been reported. Very few of these online forward triage tools have been evaluated. Evidence to decision frameworks may be of particular value in a pandemic setting where time frames are restricted, uncertainties are ubiquitous and the evidence base is changing rapidly.
Method: The objective was to develop a framework to evaluate the utility of COVID-19 online forward triage tools. The development of the online forward triage tool utility framework was conducted in three phases. The process was guided by the socio-ecological framework for adherence that states that patient (individual), societal and broader structural factors affect adherence to tool. In a further step, a pragmatic incorporation of themes on utility of online forward triage tools that emerged from our study as well as from literature was done.
Results: 7 criteria emerged; tool accessibility, reliability as an information source, medical decision-making aid, allaying fear and anxiety, health system burden reduction, onward forward transmission reduction, and systems thinking (usefulness in capacity building, planning and resource allocation e.g., tests and personal protective equipment).
Discussion and Conclusion: This framework is intended to be a starting point and a generic tool that can be adapted to other online forward triage tools beyond COVID-19. A COVID-19 online forward triage tool meeting all 7 criteria can be regarded as fit for purpose. How useful an OFTT is depends on context and purpose.
Reviewer comment 2: KEYWORDS: Keywords must be in alphabetical order. Keywords should be reviewed, so that the manuscript can be located more easily there are concepts such as telehealth that have not been included, search for MeSH descriptors.
*Author response
We thank the reviewers for the comments. We have re arranged the key words in alphabetical order. Lines 55 now read as follows;
Key words: COVID-19, evaluation framework, online forward triage tools, telehealth, utility.
Reviewer comment 3: INTRODUCTION: Line 37. They could be more explicit and indicate the technological innovations that have been developed during the pandemic and support it with previous research. Another important aspect that should be improved is that the introduction should be well organized and go from the general to the specific. If the authors address this triage system, it would be interesting for them to delve into what has been done up to now. Specify the resources and tools available.
*Author response
We thank the reviewers for highlighting this shortcoming. We have now attended to the issue. Lines 89-105 now read as follows;
The Intervention OFTT
All participants aged 18 and above that used the OFTT between 2 March and 12 May 2020 were included. COVID-19 recommendations in Switzerland changed frequently during this time period. COVID-19 testing reagents and capacity were a challenge as well as increasing the risk of a health system over load. In this pandemic phase, the Federal Office of Public Health (FOPH) recommended testing only for symptomatic patients after travel to high-risk countries (e.g., Italy and China) or symptomatic contacts of coronavirus patients. A few weeks later (as from the 20th March 2020), the recommendation was adapted to include the testing of high-risk groups (older than 65 years, pre-existing conditions, and healthcare workers). Based on how the virus spread and the availability of testing capacity, countries and risk groups were regularly adjusted. The virus spread rapidly and a universal test recommendation was made by the Federal Office of Public Health (FOPH)- on April 27th, 2020. All symptomatic individuals were eligible to test. With this recommendation, our OFTT became obsolete and was finally removed from the website on 12 May 2020, paving the way to a second generation OFTT.
Reviewer comment 3Finally, highlight the novelty of this publication since it is not clear.
*Author response
We thank the reviewers for asking this question. Kindly see lines are 91-96 for novelty
There is a paucity of data when it comes to the usefulness of these OFTTs to policy implementers, policy makers and end users. Despite the OFTT boom, very few OFTTs have been evaluated[12]. One of the reasons could be a lack of evaluative frameworks. To the best of our knowledge, there is no existing framework to assess the utility of COVID-19 OFTTs. The purpose of this manuscript is to present and propose an evaluative COVID -19 OFTT utility framework as the pandemic lingers.
Reviewer comment 5: METHODS: This section is incomplete the following subsections are missing: In phase two, it is not clear if these participants have been chosen through convenience sampling, there is no calculation of the sample size, nor inclusion and exclusion criteria.
*Author response
We thank the reviewers for these questions. We have added this info and lines146-152 now read as follows;
The database complied with Swiss laws on personal health related information collection. To minimise the barrier to OFTT use as well as for legal data protection issues, no personal data was collected within the OFTT. Further data on OFTT users was collected in a second step, from participants who gave explicit consent and provided email addresses to be contacted. A total of 560 participants consented to a follow-up survey and provided a valid e-mail address and filled in an online questionnaire (22 questions). Quantitative data was analysed in Stata® 16.1 (StataCorp, The College Station, Texas, USA).
Qualitative data lines 156-163 read as follows;
We purposefully sampled participants from those that had firstly, utilized our OFTT, secondly, had taken part in the follow-up survey and thirdly, had consented to a follow-up interview. We included participants of all age groups (quota) to ensure inclusiveness.
Sample Size
Many experts suggest saturation as central to qualitative sampling. In this study we aimed for both data saturation and rich and detailed narratives and achieved this with 19 key informants from all age groups (see Table 1).
Reviewer comment 6: The authors do not indicate any guidelines on which they have relied for the development of the manuscript in order to guarantee its quality.
*Author response
We thank the reviewers for asking this pertinent question. We followed standards for reporting qualitative research, SRQR since the themes and criteria emerged from the interviews with key stake holders
https://journals.lww.com/academicmedicine/fulltext/2014/09000/Standards_for_Reporting_Qualitative_Research__A.21.aspx
Measures to ensure trustworthiness of data: Dependability was ensured through iterative data collection and analysis and continuously adjusting our interview guide to capture newly emerging themes. Two qualitative researchers kept reflexive journals throughout the study and debriefings were held at the end of each interview. A thick description of participants, context and data collection process has been outlined to ensure transferability. Data was managed and analysed with the aid of MAXQDA2018.
Reviewer comment 7: Likewise, they do not provide references that support that the steps followed are adequate, this suggests that the manuscript may not have sufficient quality to be published. The authors explicitly self-cite in the discussion, please ignore that information.
*Author response
The study is embedded in a broader multi-phase explanatory sequential mixed methods study. Study results from both quantitative and qualitative interviews are being published separately, currently under review hence we cited them.
Reviewer comment 8: In phase 3, if it is included in the manuscript, there are aspects that are not clear, such as interviewed participants, how the interviews were carried out, where they were carried out, how the data analysis was carried out.
*Author response
We thank the reviewers for asking this important question. Answers to this question can be found in Lines 157-163
To better understand and explain the quantitative results, interviews were held with participants that further consented to a follow-up qualitative study. Video rather than face to face interviews were held in September 2020. This was a precautionary measure to protect both researchers and participants. The participants included patients (n=19) and health care providers and health authorities (n=5). To ensure inclusiveness in the patient group, participants of all age groups (purposeful and quota sampling) were included. One manuscript has been published (Michel et al., 2021). See table 1.
Reviewer comment 9: DISCUSSION. Avoid the first-person plural in the wording. The discussion of the manuscript is very short, no contract with previous research is observed. The results obtained must be contrasted with previous studies and there is only one reference in this section. The authors should carry out a deeper discussion. It would be necessary to include new future lines to investigate, after the completion of the study. It would be convenient to indicate the implications of the manuscript.
*Author response
We thank the reviewers for this suggestion: We have now done that and the discussion section now reads as follows; lines 236-274
Discussion
We propose the developed COVID-19 OFTT Utility Framework as a starting point as well as a raw guide for development OFTT utility frameworks in this and future pandemics. The seven criteria are namely; i) tool accessibility ii) tool as a reliable source of information ii) tool utility in assisting with medical decision making iv) tool potential to allay fear and anxiety v) tool potential to reduce health system burden vi) tool potential to reduce cross infection and vii) tool utility in capacity building, planning and resource allocation-systems thinking. See figure 2.
The use of OFTTs is evolving and this framework can be adapted to meet the needs of other conditions beyond COVID-19. See table 3. The views of main stakeholders in a health system, patients, health care providers and health authorities culminated in the identified criteria. The development of the OFTT utility framework was a three-phase process. The multi-phase explanatory sequential mixed methods study design enabled the team to make use of both quantitative and qualitative data facilitating a holistic picture. The socio-ecological framework for adherence states that patient (individual), societal and broader structural factors affect adherence to tool (Fig. 1 Social Ecological Framework for Nutrition and Physical Activity..., o. J.; WHO | The ecological framework, o. J.). The involvement of patients, health care providers and authorities enabled researchers to explore and incorporate individual, acceptability and equity considerations, societal, implementation , feasibility and resource implications(Fig. 1 Social Ecological Framework for Nutrition and Physical Activity..., o. J.; Stratil et al., 2020).
The “best fit” synthesis technique and the WHO -INTEGRATE COVID-19 framework guided our criteria development(Stratil et al., 2020). In a final step, a pragmatic incorporation of themes on utility of OFTTs that emerged from our quantitative and qualitative studies (Effects and Utility of an Online Forward Triage Tool during the SARS-CoV-2 Pandemic, o. J.) as well as from literature (Bauman DH., 2020; EBSCO HEALTH, 2020) was done.
Our study revealed that the OFTT has potential in reducing the health system burden, forward transmission reduction, assists in medical decision making and can act as a reliable information source. Access and socio-technical issues need to be taken into account at the point of OFTT system design and this can only be done through the involvement of all stakeholders (Zhang et al., 2020). Coronatest.ch was one of the first COVID-19 OFTTs in Switzerland. Despite the boom of OFTTs during the pandemic, there is a paucity on evaluative research of prehospital communication technology like OFTTs (Zhang et al., 2020).
All stakeholders highlighted the need for the tool to be accessible, to be a reliable source of information, to assist in medical decision making, to allay fear and anxiety to a certain extent since not all fear can be addressed, the potential to reduce the health system burden and forward transmission as well as utility in capacity building, planning and resource allocation e.g., tests and PPE-systems thinking. Guided by the “best fit” synthesis technique and the WHO -INTEGRATE COVID-19 framework the 7 criteria for our OFTT framework criteria were developed (Stratil et al., 2020). See table 3 and figure 2.
Reviewer comment 10: CONCLUSION: Check the wording, avoid the use of the first person plural. Line 176: The insights gained in our study might become obsolete as new knowledge emerges. This makes me think, if the data has been collected in 2020. And we are in 2022, it has been checked if it is obsolete. Another doubt that arises to me is that if after the elaboration of the framework it has been applied in clinical practice, that is to say, it has been piloted.
*Author response
We thank the reviewers for this question. The framework has been tested in the evaluation of a child specific COVID 19 OFTT, coronabambini and the seven criteria used to assess the utility of the tool successfully, supported by evidence and quotes from parents and guardians that used this tool. Manuscripts are currently under review at Frontiers in Public Health. See reference below;
Janet Michel, Annette Mettler, Carl Starvaggi, Christoph Aebi, Kristina Keitel, Thomas C Sauter. The utility of a pediatric COVID-19 online forward triage tool in Switzerland https://www.frontiersin.org/my-frontiers/submissions
Reviewer comment 11: REFERENCES: References must be adapted to the format of the journal.
*Author response
We thank the reviewers for highlighting this. We have adapted the references accordingly to author, date format
Reviewer 2
Open Review
(x) I would not like to sign my review report
( ) I would like to sign my review report
English language and style
( ) Extensive editing of English language and style required
( ) Moderate English changes required
(x) English language and style are fine/minor spell check required
( ) I don't feel qualified to judge about the English language and style
Yes |
Can be improved |
Must be improved |
Not applicable |
|
Does the introduction provide sufficient background and include all relevant references? |
(x) |
( ) |
( ) |
( ) |
Is the research design appropriate? |
(x) |
( ) |
( ) |
( ) |
Are the methods adequately described? |
(x) |
( ) |
( ) |
( ) |
Are the results clearly presented? |
(x) |
( ) |
( ) |
( ) |
Are the conclusions supported by the results? |
(x) |
( ) |
( ) |
( ) |
Comments and Suggestions for Authors
*Author response
We thank the reviewers for the encouraging comments.
Reviewer comment 1: In this paper the authors describe the utility of triage tools in the setting of COVID19 pandemy. The paper is interesting and well written. I have just minor comments:
- the term OFTTs should also be defined in the abstract
- the discussion section lacks of a better explanation of a suitable future applications of this kind of tools beyond the pandemy
*Author response
We thank the authors for highlighting this. We have attended to the issue. See abstract at the top
- x) I would not like to sign my review report
( ) I would like to sign my review report
English language and style
( ) Extensive editing of English language and style required
( ) Moderate English changes required
(x) English language and style are fine/minor spell check required
( ) I don't feel qualified to judge about the English language and style
Yes |
Can be improved |
Must be improved |
Not applicable |
|
Does the introduction provide sufficient background and include all relevant references? |
(x) |
( ) |
( ) |
( ) |
Is the research design appropriate? |
( ) |
(x) |
( ) |
( ) |
Are the methods adequately described? |
( ) |
( ) |
(x) |
( ) |
Are the results clearly presented? |
( ) |
( ) |
(x) |
( ) |
Are the conclusions supported by the results? |
( ) |
( ) |
(x) |
( ) |
Comments and Suggestions for Authors
Reviewer comment: This study attempts to develop an online forward triage tool utility framework for COVID-19. A case study was conducted. This is a meaningful study. Here are some comments for the authors to consider.
1. According to Line 73-77 and Table 2, the methods and results are from existing literature. What are new in the current study?
*Author response
We thank the reviewers for asking this question. We have adapted the methods and results sections and also included the intervention description. See above
Reviewer comment: It unclear how the 7 criteria and their relationships are determined in this study. The authors mention that the framework was developed by brain storming and discussions with research team based on literature... (p. 3). It is suggested to report these literatures and how each literature is used. Also, are there any scientific approaches were used for guiding the brain storming and discussion with the research team?
*Author response
We thank the reviewers for asking this important question. See Lines 130-1391
Framework development
The development of the OFTT utility framework was conducted in three phases. The process was guided by the socio-ecological framework for adherence that states that patient (individual), societal and broader structural factors affect adherence to tool (Fig. 1 Social Ecological Framework for Nutrition and Physical Activity..., o. J.; WHO | The ecological framework, o. J.). The “best fit” synthesis technique and the WHO -INTEGRATE COVID-19 framework guided our criteria development (Stratil et al., 2020). In a further step, a pragmatic incorporation of themes on utility of OFTTs that emerged from our study (Effects and Utility of an Online Forward Triage Tool during the SARS-CoV-2 Pandemic, o. J.) as well as from literature (Bauman DH., 2020; EBSCO HEALTH, 2020) . See table 1.
Reviewer comment: It should be reported that how the points assigned to each criterion in table 3 are scientifically determined.
*Author response
We thank the reviewers for this question lines 213-231
How to apply the framework
This framework is intended to be a starting point and a generic tool that can be adapted to other OFTTS beyond COVID-19. The framework development was guided by the “best fit” synthesis technique and the WHO -INTEGRATE COVID-19 framework (Stratil et al., 2020).
The seven criteria are; i) tool accessibility ii) tool as a reliable source of information ii) tool utility in assisting with medical decision making iv) tool potential to allay fear and anxiety v) tool potential to reduce health system burden vi) tool potential to reduce cross infection vii) tool utility in capacity building, planning and resource allocation-systems thinking.
See table 2. The 7 criteria can be adapted to suit a specific purpose e.g., addition of more criteria to suit the purpose of OFTT e.g., in acute or chronic conditions. See figure 2. How useful an OFTT is can be assessed by how the tool meets the 7 criteria. A COVID-19 tool meeting all 7 criteria is fit for purpose. A COVID-19 tool meeting 5-6 criteria is good and a COVID-19 tool meeting 4 and less criteria is insufficient. The number of criteria is determined by purpose of OFTT. A chronic pain OFTT might have more or less criteria as compared to a urinary tract infection OFTT. An OFTT whose main purpose is to provide information might need to fulfil criteria 1 tool accessibility and criteria 2 a reliable and comprehensive information source. There is no one size fits all. The identified criteria are meant primarily for COVID-19 OFTTs. The context and purpose of tool will dictate the relevant criteria to be fulfilled. See table 3.
Reviewer comment. The contents in Section 2.5-2.7 do not seem to be complete.
*Author response
We thank the reviewers. We have attended to all sections and made adjustments accordingly
Reviewer 2 Report
In this paper the authors describe the utility of triage tools in the setting of COVID19 pandemy. The paper is interesting and well written. I have just minor comments:
- the term OFTTs should also be defined in the abstract
- the discussion section lacks of a better explanation of a suitable future applications of this kind of tools beyond the pandemy
Author Response

(The authors gave the same response as above.)

Reviewer 3 Report
This study attempts to develop an online forward triage tool utility framework for COVID-19. A case study was conducted. This is a meaningful study. Here are some comments for the authors to consider.
1. According to Line 73-77 and Table 2, the methods and results are from existing literature. What are new in the current study?
2. It unclear how the 7 criteria and their relationships are determined in this study. The authors mention that the framework was developed by brain storming and discussions with research team based on literature... (p. 3). It is suggested to report these literatures and how each literature is used. Also, are there any scientific approaches were used for guiding the brain storming and discussion with the research team?
3. It should be reported that how the points assigned to each criterion in table 3 are scientifically determined.
4. The contents in Section 2.5-2.7 do not seem to be complete.
Author Response

(The authors gave the same response as above.)

Round 2
Reviewer 1 Report
All the recommendations have been addressed
Author Response
Inselspital
Freiburgstrasse 18
3010 Bern
14 April 2022
Editor IJERPH
Dear Editor
RE: A Utility Framework for COVID-19 Online Forward Triage Tools: A Swiss Telehealth case study
We thank the reviewers for their invaluable comments on our manuscript; “A Utility Framework for COVID-19 Online Forward Triage Tools: A Swiss Telehealth case study.” We have carefully taken all the reviewers’ comments into account and are pleased to submit a revised version herewith. We indicate below how we have changed the manuscript and respond to the reviewers’ comments item by item.
Sincerely,
Janet Michel, RN, BA Cur, MPH, PHD
Annette Mettler, MD
Thomas C. Sauter, MD, MME, Prof
Reviewer 1
(x) I would not like to sign my review report
( ) I would like to sign my review report
English language and style
( ) Extensive editing of English language and style required
( ) Moderate English changes required
(x) English language and style are fine/minor spell check required
( ) I don't feel qualified to judge about the English language and style
Yes |
Can be improved |
Must be improved |
Not applicable |
|
Does the introduction provide sufficient background and include all relevant references? |
(x) |
( ) |
( ) |
( ) |
Is the research design appropriate? |
(x) |
( ) |
( ) |
( ) |
Are the methods adequately described? |
(x) |
( ) |
( ) |
( ) |
Are the results clearly presented? |
(x) |
( ) |
( ) |
( ) |
Are the conclusions supported by the results? |
(x) |
( ) |
( ) |
( ) |
Comments and Suggestions for Authors
All the recommendations have been addressed
Submission Date
09 March 2022
Date of this review
10 Apr 2022 08:22:51
*Author response
We thank the reviewer for these encouraging comments. We have also incorporated the suggested references. Lines 371-374 now read as follows;
A telemedicine monitoring framework to assist infected patients remotely was developed elsewhere (Battineni, Pallotta, et al., 2021). The authors argue that telemedicine has the potential to a health care source in pandemics(Battineni, Pallotta, et al., 2021).
Lines 330-353
Our study revealed that the OFTT has potential in reducing the health system burden, forward transmission reduction, assists in medical decision making and can act as a reliable information source concurring with a studies elsewhere (Battineni, Nittari, et al., 2021).
Reviewer 2
(x) I would not like to sign my review report
( ) I would like to sign my review report
English language and style
( ) Extensive editing of English language and style required
( ) Moderate English changes required
(x) English language and style are fine/minor spell check required
( ) I don't feel qualified to judge about the English language and style
Yes |
Can be improved |
Must be improved |
Not applicable |
|
Does the introduction provide sufficient background and include all relevant references? |
(x) |
( ) |
( ) |
( ) |
Is the research design appropriate? |
(x) |
( ) |
( ) |
( ) |
Are the methods adequately described? |
(x) |
( ) |
( ) |
( ) |
Are the results clearly presented? |
(x) |
( ) |
( ) |
( ) |
Are the conclusions supported by the results? |
(x) |
( ) |
( ) |
( ) |
Comments and Suggestions for Authors
The revised manuscript shows major improvements. There are still some minor issues.
- Table 1 does not look neat as the font size for the first line of Table 1 (i.e., stage 1, 2, 3) is big.
*Author response
The font has been adapted to MDPI format. All font is now in Palatino Linotype
- In the section 3. Results, there is only one subsection 3.1.
*We thank the reviewers for these comments. Yes we agree with the reviewer that there is one subsection in the results section. The reason is that the purpose of the manuscript is to present the framework, hence we dwelt on this more. We however have links to papers, some under review for anyone interested in the detailed results.
- The font size for some words is different from others in Line 234-242.
*Author response
The font in the whole manuscript is now in Palatino Linotype
Submission Date
09 March 2022
Date of this review
13 Apr 2022 11:28:50
Reviewer 3 Report
The revised manuscript shows major improvements. There are still some minor issues.
1. Table 1 does not look neat as the font size for the first line of Table 1 (i.e., stage 1, 2, 3) is big.
2. In the section 3. Results, there is only one subsection 3.1.
3. The font size for some words is different from others in Line 234-242.
Author Response

(The authors gave the same response as above.)
